# First detection of Koi herpesvirus disease (KHVD) in Garmian, Kurdistan region of Iraq: A clinical and molecular study

Diyar Akbar Hasan Al-Jaf[1]*, Shaymaa Abdulaziz Nawokas[2], Muhammad Majed Mardoukhi[3]

1 College of Medicine, University of Garmian, Kalar, Iraq, 2 Biology Department, College of Education, University of Garmian, Kalar, Iraq, 3 Kalar Private Veterinary Clinic, Kalar, Iraq

* diyar.akbar@garmian.edu.krd

**Data Availability Statement:** All relevant data are within the manuscript and its Supporting Information files.

## Abstract

### Introduction

Koi herpesvirus disease (KHVD) is attributed to cyprinid herpesvirus-3 (CyHV-3) and predominantly affects common carp and ornamental koi carp (Cyprinus carpio). This viral infection leads to substantial morbidity and mortality among these fish species. This study aimed to confirm the presence of KHVD in the Kurdistan region of Iraq by employing clinical and optimized molecular assays on fish populations experiencing high mortality among common carp in carp farms.

### Methodology

The present research was conducted in the Kalar district, situated at the heart of Garmian province in Iraqi Kurdistan. four samples from common carp fish farms were received by our laboratory. These samples specifically displaying clinical signs associated with koi herpesvirus (KHV) infection, were subjected to clinical examinations, and PCR assay in addition to sequence analysis.

### Results

The results of the current study revealed that the observed clinical signs, particularly gill necrosis, skin lesions, and sunken eyes, closely resembled the clinical signs of KHVD in common carp fish. In addition, PCR, nested PCR, and sequence analysis assay detected appropriate DNA fragments of the CyHV-3 major capsid protein gene confirming the first detection of KHVD in common carp fish in the Kurdistan region of Iraq.

### Conclusion

In this study, the results confirm the detection of KHVD in the Kurdistan region, Iraq, for the first time. This study revealed that CyHV-3 was responsible for KHVD-related signs and symptoms. Based on these results, it is strongly recommended that comprehensive studies be initiated to investigate the prevalence and distribution of CyHV-3.

**Funding:** The author(s) received no specific funding for this work.

## 1. Introduction

Koi herpesvirus disease (KHVD) is a disease caused by cyprinid herpesvirus-3 (CyHV-3), primarily affecting common carp and ornamental koi carp (Cyprinus carpio) in Europe and Asia. It results in significant morbidity and mortality in these fish and is characterized by clinical signs such as white patches, skin hemorrhages, lethargy, reduced appetite, sunken eyes, enlargement of the spleen and kidney, and gill necrosis [1].

CyHV3, belonging to the Alloherpesviridae family and the genus Cyprinivirus, possesses the longest genome within the Herpesvirales order, consisting of a double-stranded DNA approximately 295 kbp in size [2, 3]. Using the genes responsible for viral DNA polymerase and the ATPase subunit of terminase as a reference, CyHV-3 is presently categorized alongside CyHV-1 and CyHV-2 in the Cyprinivirus genus within the Alloherpesviridae family. The mature viral particles are 170 to 200 nm in diameter. CyHV-3 can be detected by multiple PCR-based methods and by enzyme-linked immunosorbent assay [1, 4].

KHVD has decimated major carp populations in various countries such as Israel, Indonesia, Taiwan, Japan, Germany, Canada, and the United States. It has been a notifiable disease in Germany since 2005 and by the World Organization for Animal Health since 2007 and has spread to most regions of the world [5, 6].

In 1998, initial large-scale deaths of both common and koi carp were documented in Israel and the United States [7]. However, examinations of archived samples revealed that the virus has been present in wild common carp since 1996 in the United Kingdom [8]. Shortly after the initial report, outbreaks of Cyprinid herpesvirus-3 (CyHV-3) were observed in various countries across Europe, Asia, and Africa. Currently, CyHV-3 has been identified worldwide, except in South America, Australia, and Northern Africa [5]. Globally, CyHV-3 has led to substantial financial and economic losses in the Koi and common carp aquaculture industries. In the countries neighboring Iraq, the first detection of Koi herpesvirus (KHV) infection in koi occurred in Iran in 2015, as documented through clinical Histopathological and molecular studies [9].

In late October 2018, Iraq reported the first documented case of a Koi herpesvirus (KHV) outbreak. The incident impacted the carp farming industry in the central Euphrates region of Iraq, resulting in the deaths of several fish. The virus responsible for this outbreak was identified as Cyprinid herpesvirus 3 (CyHV-3), the causative agent of KHVD. This marked the first known occurrence of KHV in Iraq and had a substantial effect on the country's carp farming sector [10].

Aquaculture practices in Iraq emerged in the mid-to-late 20th century, with the introduction of the common carp (Cyprinus carpio) in 1955 at the Al-Zaafaraniya fish farm in Baghdad [11]. It is thriving in Iraq, primarily relying on the Tigris and Euphrates rivers for farming because of the country's limited coastline along the Gulf [12]. These freshwater fish farms exist in both the public and private sectors, with their concentration mainly in the central and southern regions of Iraq. The production of freshwater fish in Iraq is predominantly centered on common carp (Cyprinus carpio carpio), although there is also limited cultivation of grass carp (Ctenopharyngodon idella) and silver carp (Hypophthalmichthys molitrix). In 2015, the total freshwater carp production in Iraq was estimated to be approximately 30,000 tons [13, 14].

The common carp industry in the Kurdistan region of Iraq developed only during the last twenty years or so, and now spread to many farms in the region [15]. It is a flourishing sector that contributes to economic growth. Common carp, a popular food fish, is cultivated as a cost-effective means of protein production. The industry has experienced significant growth recently, driven by an increasing demand for fish and governmental support. Numerous fish

farms in rural areas near water sources, such as rivers and lakes, focus on raising common carp in ponds or tanks. These fish are fed a combination of commercial pellets and natural foods. Common carp's resilience, ability to tolerate poor water quality, and ability to withstand high temperatures make them well-suited for aquaculture in the Kurdistan Region, particularly in areas with limited water resources [16]. currently no documented evidence of KHVD in the Kurdistan Region of Iraq.

The study could address the gap in understanding the prevalence and distribution of this virus in the region. The study aimed to confirm the presence of Koi herpesvirus Disease (KHVD) in the Kurdistan Region by employing clinical and molecular assays on fish populations experiencing high mortality among common carp in carp farms Iraqi Kurdistan region.

## 2. Materials and methods

### 2.1. Study area

The present research was conducted in the Kalar district, situated at the heart of Garmian province in the Kurdistan region of Iraqi, positioned around 140 km southeast of Sulaymaniyah and 30 km from the Iranian border, the district encompasses a geographical area with a population of approximately 250,000 residents.

### 2.2. Sample collection

Our laboratory received samples from four farms of common carp fish located in Kalar district: two samples in December 2022, one sample in June 2023, and another sample in October 2023. Fish exhibiting clinical signs of koi herpesvirus (KHV) infection, including lethargy, anorexia, increased mucus secretion, skin lesions, gill discoloration and necrosis, and sunken eyes, were targeted for sampling. Since the fish were deceased, note that no anesthesia or analgesia was administered to minimize suffering.

### 2.3. DNA extraction

The extraction of genomic DNA from fish tissues was conducted in the following manner: the skin, gills, liver, kidney, heart, spleen, and brain tissues were finely sliced, followed by grinding and homogenization using a mortar and pestle with the addition of liquid nitrogen. The extraction of genomic viral nucleic acid from these tissues was accomplished using the AddPrep Viral Nucleic Acid Extraction Kit (Add Bio Inc, Korea) in accordance with the manufacturer's guidelines.

### 2.4. Primer design

Two sets of primers were designed for the specific detection of CyHV-3 in tissues of suspected samples. Previously, the primers were checked for homology and found to be specific to CyHV-3. The external primer pair amplified a 529 bp fragment of CyHV-3 DNA, and the internal primer pair amplified a 379 bp fragment of CyHV-3 DNA. These primers were previously described by El-Matbouli *et al.* (2007; GenBank Accession number AY939864) [17]. The external primer pair (sense: 5′-ACCAACTTTAGCACGGACGAC-3′; antisense: 5′-ACTTGACCAGGTACAGCAGC-3′) and the internal primer pair for nested PCR (sense: 5′-CTTTAGCACGGACGACTTTGGC-3′; antisense: 5′-GTAGACGATGGACAGGGAG-3′). We used the primers introduced by Kocher et al. (1989) and Boakye et al. (1999) for amplification of a portion (358 bp) of CytB gene of vertebrate mtDNA as a positive control [18, 19].

## 2.5. PCR

PCR amplification was performed using 20 μl total volume as follows: the reactions mixture was prepared by mixing 10 μl Add start master mix 2 x (Add Bio Inc, Korea) with 1 μl of each external primer (5 pM) and 5 μl genomic DNA in addition to 3 μl of Nuclease-Free Water. The reactions were run in a thermal cycler (Applied Biosystems 2720, Foster City, CA 94404 USA) by initial denaturation at 95˚C for 5 minutes; followed by 40 cycles of 95˚C for 30 seconds, annealing temperature at 60˚C for 45 seconds and 72˚C for 30 seconds; and finally extended at 72˚C for 5 minutes. The specific PCR amplification protocol used for nested PCR was the same as the protocol used for the first PCR but with the internal set of primer. The only difference was that the genomic DNA template for the nested PCR was the amplification product of the first PCR [17].

The amplified PCR products obtained from the first PCR and the nested PCR were separated using gel electrophoresis on a 1.5% agarose gel. The electrophoresis buffer used was Tris-Borate-EDTA-Na$_2$ (TBE) buffer. The amplified PCR products were visualized by staining the gel with 5 μl ethidium bromide (0.5 μg/mL). To determine the molecular size of the PCR and nested PCR amplicons, a 100 bp DNA ladder from GeneDireX, Inc, was used as a reference.

## 2.6. Ethical approval

The ethical committee at the Research Centre, University of Garmian granted the necessary ethical approval for the study.

# 3. Results

## 3.1. Clinical findings

Upon clinical examination, changes in gill appearance were the most prominent clinical finding. The gills of the affected fish exhibited gross changes, including varying degrees of discoloration and necrosis, accompanied by an excessive production of slimy mucus The fish's eyes were noticeably sunken. Skin ulcerations were also among prominent clinical finding, with lesions varying in size and scattered throughout the body. These ulcers merged and could reach up to 2cm in diameter (Fig 1).

## 3.2. PCR detection

Positive results were detected when fragments of the appropriate size of 529 bp for PCR and 379 bp for nested PCR of the CyHV-3 major capsid protein (MCP) gene (accession number AY939864) [20] was amplified from the CyHV-3 suspected samples (Fig 2).

## 3.3. Sequence analysis

Partial sequence analysis and phylogenetic tree using Mega11 revealed that the major capsid protein (MCP) gene, was identical to that of several KHV strains in the GenBank database: examples TUMST1 (Japan, accession number AP008984) and KHV-I (Israel, DQ177346) (Fig 3).

# 4. Discussion

The study aimed to verify the existence of Koi herpesvirus Disease (KHVD) in the Kurdistan Region by utilizing clinical and molecular assays including sequence analysis on fish populations experiencing mass mortality among common carp in carp farms within the Iraqi Kurdistan region, with confirmation through sequence analysis. We describe the first detection of

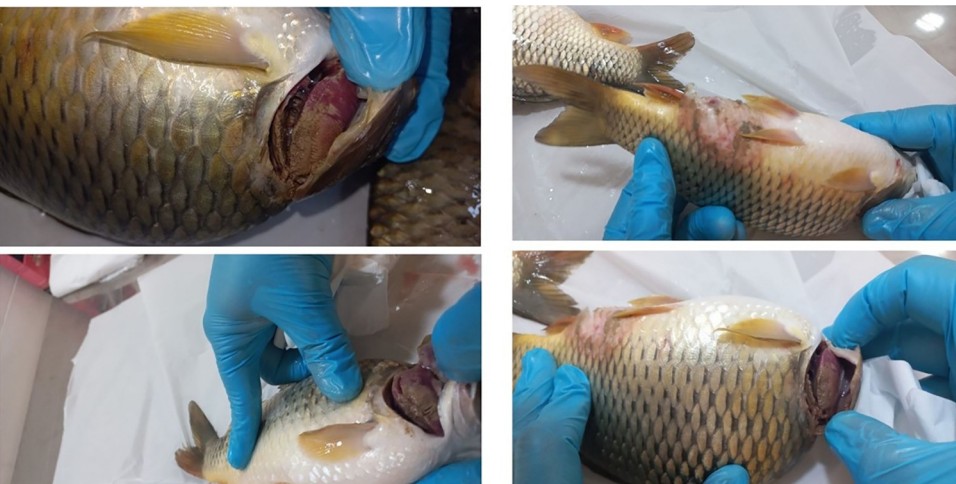

**Fig 1. Fish, carp, whole fish.** Multiple variably sized, irregular skin ulcers gill discoloration and necrosis.

KHV in common carp fish in the Iraqi Kurdistan region because no study addressed KHVD before this region, but in Iraq, KHVD outbreak that started in late October 2018 [21]. The Disease occurred when the water temperature was optimal for the growth and replication of CyHV-3, which is between 18–28°C but no mortality was observed for CyHV-3-exposed fish at 13°C [22]. KHVD was reported by four countries during the period from 1 January 2018 to 20 March 2019. The disease was first reported in Romania in June 2018, and then in Canada in September 2018, Italy in December 2018, and Iraq in January 2019. As of 20 March 2019, there are still ongoing KHVD outbreaks in Romania, Italy, and Iraq [23].

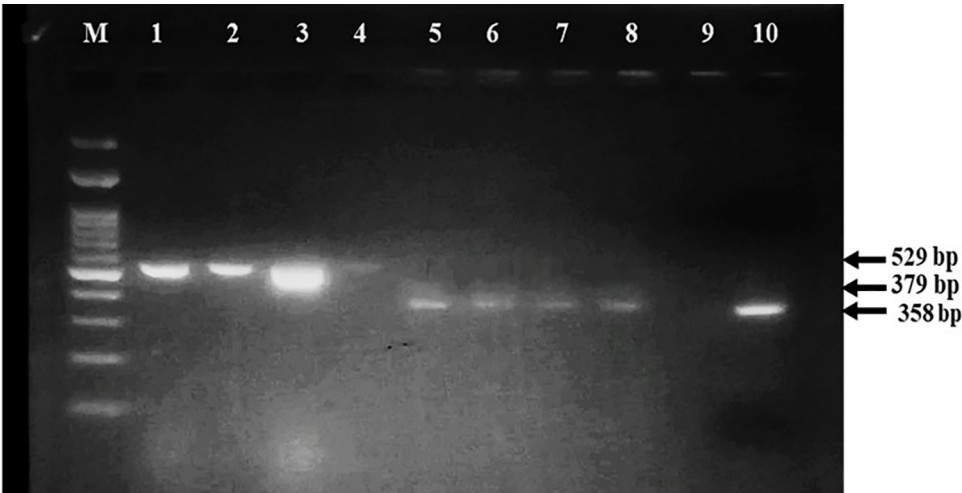

**Fig 2. Agarose gel electrophoresis Cyprinid herpesvirus-3 (CyHV-3) PCR and nested PCR product utilizing specific external and internal primer sets of the CyHV-3 major capsid protein gene (529 and 379 bp respectively) for samples from four fish farms.** Separation is on a 1.5% agarose gel stained with ethidium bromide. Lanes are M: 100-bp DNA ladder, lanes1-4: Expected amplicon size of 529 bp in the four samples specific for major capsid protein gene of CyHV-3, Lane 5–8: show expected nested PCR amplicon size of 379 bp in the four samples, Lane 9: negative PCR control, Lane: Positive control, PCR amplifications of vertebrate cytB sequences (358 bp).

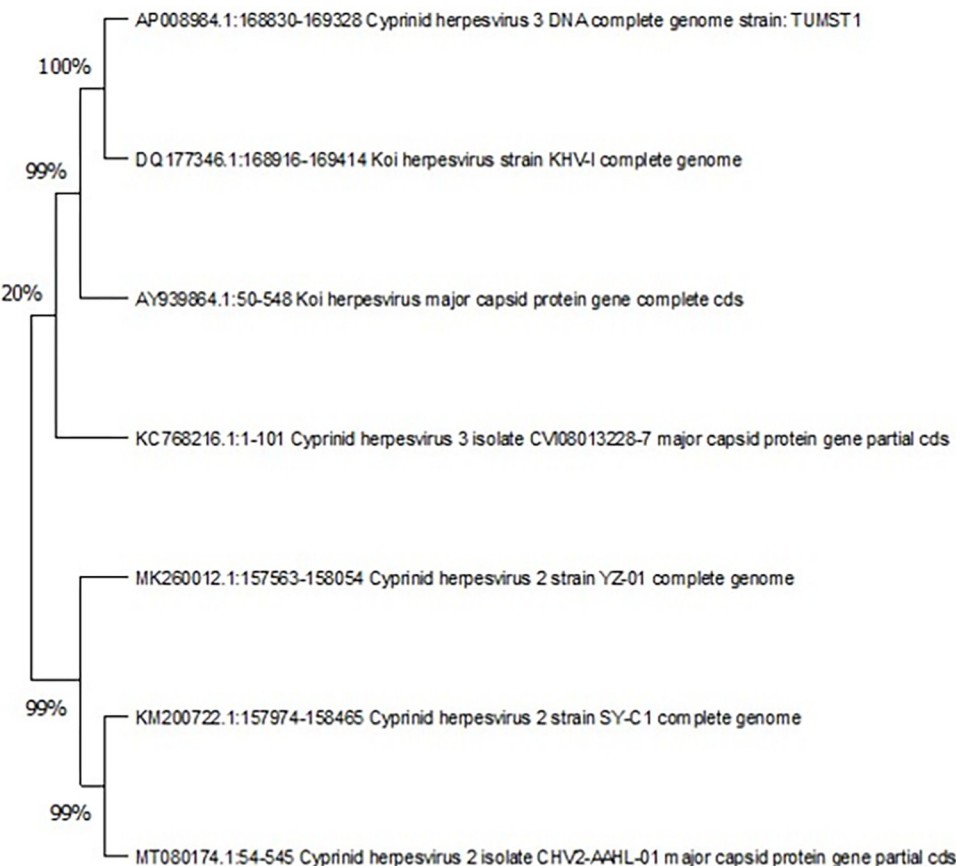

**Fig 3. Phylogenetic analysis of Kurdistan region isolates of Koi herpesvirus (KHV) was conducted, comparing them with seven foreign CyHV-3 isolates based on the partial nucleotide sequences of the MCP gene.** Initially, the ClustalW tool was employed to align nucleotide sequences, followed by construction of the Phylogenetic tree using the neighbor-joining method through the MEGA 11 program.

Clinical signs of KHVD in fish include pale and necrotic gills, enophthalmus, and excessive secretion of mucus [1]. Similar clinical signs were observed in the present study, especially gill necrosis, skin lesions and sunken eye.

For rapid and accurate diagnosis of acute KHV infections, PCR-based techniques have been shown to be the most sensitive diagnostic tools for KHV detection, as reported by Haenen et al. [24, 25]. Our laboratory used the CyHV-3 major capsid protein gene-based PCR protocol with external and internal primers to amplify 529 and 379 bp fragments from CyHV-3 DNA. The sensitivity of this PCR protocol has been well addressed before by El-Matbouli et al. [17].

The results confirmed that KHV behand the KHVD-related clinical signs in four fish samples tested by the mentioned PCR protocol. previously no microbiological and molecular study was conducted on viral disease in common carp fish farms in the region, while most studies concentrated on bacterial and protozoal disease [26–28]. Outbreaks of the disease in this region have occurred in different period of the years without regard to temperature, may be due to the fact the disease new to region and it is first exposure to CyHV-3. The presence of secondary parasitic infections along with CyHV-3 in diseased carp can complicate the pathogenesis of the disease [21]. These secondary infections are prevalent in CyHV-3-affected carp and can worsen the clinical symptoms and increase mortality rates [29].

The swift and widespread of KHV is likely attributed to a variety of factors, such as the international trade in ornamental and aquarium fish, deficiencies in historical diagnostic methods, and the absence of regulations for controlling and preventing the virus's introduction [30].

The precise origin and mode of introduction of CyHV-3 to carp farms in the Kurdistan region of Iraq remain undetermined. However, based on existing research, several scenarios can be hypothesized. The most probable pathways are the movement of live fish between fish farms in this region and the rest of Iraq or the importation of carp. In addition, wild non-cyprinid fish could pose a potential risk factor for CyHV-3 infection [31].

Another reason that makes tracing the source of the disease challenging is the ability of KHV to establish latent infections. This means that the virus can maintain its DNA in fish hosts without causing any visible symptoms or signs of disease. Latent KHV infections have been reported, particularly in fish presumed to have been previously exposed to KHV [32]. The virus can reemerge from its dormant state when environmental factors such as temperature fluctuations create favorable conditions for its growth [33].

To prevent the spread of KHV, early detection is paramount. PCR, a powerful technique capable of detecting minute viral quantities, is deemed effective for identifying KHV latency in common carp [34]. Therefore, this study employed previously optimized PCR assay [17] to assess the presence of KHV, particularly in common carp from Kalar district, Kurdistan.

## 5. Conclusions

It is suggested that in this study, the results confirm the detection of KHVD in Kurdistan, Iraq, for the first time in common carp (*Cyprinus carpio*) freshwater farms by Research study. Clinical and molecular examination revealed that CyHV-3 was responsible for Koi herpesvirus disease (KHVD) related signs, symptoms, and mortality.

## 6. Recommendation

Based on this result, it is strongly recommended to initiate comprehensive studies aimed at assessing the prevalence and distribution of CyHV-3 in diverse carp farms across the Kurdistan region. Furthermore, conducting genetic analyses of CyHV-3 isolates in Kurdistan will contribute to the development of efficient strategies for managing and preventing the spread of the disease in the region.

## Supporting information

**S1 Fig.**
(TIF)

**S2 Fig.**
(TIF)

**S3 Fig.**
(TIF)

**S4 Fig.**
(TIF)

**S1 File.**
(DOCX)

## Author Contributions

**Conceptualization:** Muhammad Majed Mardoukhi.

**Funding acquisition:** Shaymaa Abdulaziz Nawokas.

**Methodology:** Diyar Akbar Hasan Al-Jaf, Muhammad Majed Mardoukhi.

**Project administration:** Diyar Akbar Hasan Al-Jaf.

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
