## [Decision Letter · Decision Letter 0]

12 Apr 2024

PONE-D-24-08826First Detection of Koi herpesvirus Disease (KHVD) in Garmian, Kurdistan Region of Iraq: a clinical and molecular studyPLOS ONE

Dear Dr. LAST_NAME%

Thank you for submitting your manuscript to PLOS ONE. After careful consideration, we feel that it has merit but does not fully meet PLOS ONE’s publication criteria as it currently stands. Therefore, we invite you to submit a revised version of the manuscript that addresses the points raised during the review process.

We look forward to receiving your revised manuscript.

Kind regards,

Academic Editor

PLOS ONE

Journal Requirements:

2. To comply with PLOS ONE submissions requirements, in your Methods section, please provide additional information regarding the experiments involving animals and ensure you have included details on (1) methods of sacrifice, (2) methods of anesthesia and/or analgesia, and (3) efforts to alleviate suffering

4. We note that your Data Availability Statement is currently as follows: "All relevant data are within the manuscript and its Supporting Information files"

Additional Editor Comments :

The authors are advised to incorporate the suggestions of reviewers. Include some introductory part about the loss caused by diseases in aquaculture.

Kindly refer the following citations/manuscripts

1. Bhat, R., Tandel, R. and Pandey, P.K., 2022. Alternatives to antibiotics for combating the antimicrobial resistance in aquaculture. Indian J Animal Health, 61, pp.01-18.

2. Bhat, R.A.H., Rehman, S., Tandel, R.S., Dash, P., Bhandari, A., Ganie, P.A., Shah, T.K., Pant, K., Yousuf, D.J., Bhat, I.A. and Chandra, S., 2021. Immunomodulatory and Antimicrobial potential of ethanolic extract of Himalayan Myrica esculanta in Oncorhynchus mykiss: Molecular modelling with Aeromonas hydrophila functional proteins. Aquaculture, 533, p.736213.

3. Bhat, R.A.H. and Altinok, I., 2023. Antimicrobial Resistance (AMR) and Alternative Strategies for Combating AMR in Aquaculture. Turkish Journal of Fisheries and Aquatic Sciences, 23(11).

4. Bhat, R.A.H., Khangembam, V.C., Pant, V., Tandel, R.S., Pandey, P.K. and Thakuria, D., 2024. Antibacterial activity of a short de novo designed peptide against fish bacterial pathogens. Amino Acids, 56(1), pp.1-13.

Reviewers' comments:

Reviewer's Responses to Questions

**Comments to the Author**

1. Is the manuscript technically sound, and do the data support the conclusions?

Reviewer #1: Yes

Reviewer #2: Yes

2. Has the statistical analysis been performed appropriately and rigorously? 

Reviewer #1: Yes

Reviewer #2: N/A

3. Have the authors made all data underlying the findings in their manuscript fully available?

Reviewer #1: Yes

Reviewer #2: Yes

4. Is the manuscript presented in an intelligible fashion and written in standard English?

Reviewer #1: Yes

Reviewer #2: Yes

5. Review Comments to the Author

Reviewer #1: The reviewer comments and the requested revisions to the author are represented in the attached file. The performing the requested revisions make the manuscript worthy publication. You are doing very well. Hope the success for the research team

Reviewer #2: Dear Authors,

you prepared an interesting manuscript, showing the prevalence of KHV in Garmian, Iran. Your work is clearly structured and comprehensible.

Some minor comments:

- it is Cyprinid Herpesvirus-3

- it is koi not koi carp, the Japanese word "koi" stands for carp

- it is longest, not lengthiest

- please include your sequence alignment to your sequence analysis

6. PLOS authors have the option to publish the peer review history of their article (what does this mean?). If published, this will include your full peer review and any attached files.

Reviewer #1: No

Reviewer #2: No

---

## [Author Response · Author response to Decision Letter 0]

20 Apr 2024

I have replies to all comment by the reviewers and the editor in a PDF uploaded as "response to the reviewer"

---

## [Editor Report · Decision Letter 1]

25 Apr 2024

First Detection of Koi herpesvirus Disease (KHVD) in Garmian, Kurdistan Region of Iraq: a clinical and molecular study

PONE-D-24-08826R1

Dear Dr. Diyar Akbar

We’re pleased to inform you that your manuscript has been judged scientifically suitable for publication and will be formally accepted for publication once it meets all outstanding technical requirements.

Kind regards,

RAJA AADIL HUSSAIN BHAT

Academic Editor

PLOS ONE

---

## [Editor Report · Acceptance letter]

10 May 2024

PONE-D-24-08826R1 

PLOS ONE

Dear Dr. Akbar, 

I'm pleased to inform you that your manuscript has been deemed suitable for publication in PLOS ONE. Congratulations! Your manuscript is now being handed over to our production team.

Kind regards, 

on behalf of

Dr. RAJA AADIL HUSSAIN BHAT 

Academic Editor

PLOS ONE